# Deficiency of Polη in *Saccharomyces cerevisiae* reveals the impact of transcription on damage-induced cohesion

**Pei-Shang Wu**, **Jan Grosser**[Ѡ], **Donald P. Cameron**[Ѡ], **Laura Baranello**, **Lena Ström** *

Karolinska Institutet, Department of Cell and Molecular Biology, Stockholm, Sweden

Ѡ These authors contributed equally to this work.
* lena.strom@ki.se

**Data Availability Statement:** The authors confirm that all data underlying the findings are fully available without restriction. RNA-seq data are found at: Gene Expression Omnibus 163287

## Abstract

The structural maintenance of chromosome (SMC) complex cohesin mediates sister chromatid cohesion established during replication, and damage-induced cohesion formed in response to DSBs post-replication. The translesion synthesis polymerase Polη is required for damage-induced cohesion through a hitherto unknown mechanism. Since Polη is functionally associated with transcription, and transcription triggers *de novo* cohesion in *Schizosaccharomyces pombe*, we hypothesized that transcription facilitates damage-induced cohesion in *Saccharomyces cerevisiae*. Here, we show dysregulated transcriptional profiles in the Polη null mutant (*rad30Δ*), where genes involved in chromatin assembly and positive transcription regulation were downregulated. In addition, chromatin association of RNA polymerase II was reduced at promoters and coding regions in *rad30Δ* compared to WT cells, while occupancy of the H2A.Z variant (Htz1) at promoters was increased in *rad30Δ* cells. Perturbing histone exchange at promoters inactivated damage-induced cohesion, similarly to deletion of the *RAD30* gene. Conversely, altering regulation of transcription elongation suppressed the deficient damage-induced cohesion in *rad30Δ* cells. Furthermore, transcription inhibition negatively affected formation of damage-induced cohesion. These results indicate that the transcriptional deregulation of the Polη null mutant is connected with its reduced capacity to establish damage-induced cohesion. This also suggests a linkage between regulation of transcription and formation of damage-induced cohesion after replication.

## Author summary

The cohesin complex dynamically associates with chromosomes and holds sister chromatids together through cohesion established during replication. This ensures faithful chromosome segregation at anaphase. In budding yeast, DNA double strand breaks also trigger sister chromatid cohesion after replication. This so-called damage-induced cohesion is formed both close to the breaks, and genome-wide on undamaged chromosomes. The translesion synthesis polymerase eta (Polη) is specifically required for genome wide damage-induced cohesion. Although Polη is well characterized for its function in

(https://www.ncbi.nlm.nih.gov/geo/query/acc.cgi?
acc=GSE163287) Scc1-ChIP-seq data are found at:
Gene Expression Omnibus 42655 (https://www.
ncbi.nlm.nih.gov/geo/query/acc.cgi?acc=
GSE42655) Polη-ChIP-sequencing data are found
at Gene Expression Omnibus 179539 (https://
www.ncbi.nlm.nih.gov/geo/query/acc.cgi?acc=
GSE179539) Numerical data underlying graphs are
found in S4 Data, and a statistics summary is
found in S5 Data. Uncropped images for the
immunofluorescence images in S7D Fig are
deposited in figshare and available at 10.6084/m9.
figshare.15173295.

**Funding:** LS was supported by: Vetenskapsrådet,
Swedish Research Council (2016-02206), https://
www.vr.se, Cancerfonden, Swedish Cancer Society
(2016/554, 2019/410), https://www.cancerfonden.
se, Magnus Bergvalls stiftelse, Bergvall Foundation
(2016-01868, 2017-02287), http://www.
magnbergvallsstiftelse.nu Karolinska Institutet,
PhD student financing program (3-1818/2013),
https://staff.ki.se/kid-funding LB was supported by:
Cancerfonden, Swedish Cancer Society (2018/
760), https://www.cancerfonden.se, Karolinska
Institutet, PhD student financing program (2-5586/
2017), https://staff.ki.se/kid-funding Knut och Alice
Wallenbergs stiftelse, Knut and Alice Wallenberg
foundation (2016.0161), https://kaw.wallenberg.
org Vetenskapsrådet, Swedish Research Council
(2016-02610), https://www.vr.se The
computations and data storage were enabled by
resources in project [SNIC 2018/8-390] provided
by the Swedish National Infrastructure for
Computing (SNIC) at UPPMAX, partially funded by
the Swedish Research Council through grant
agreement no. 2018-05973. The funders had no
role in study design, data collection and analysis,
decision to publish, or preparation of the
manuscript.

**Competing interests:** The authors have declared
that no competing interests exist.

bypassing ultraviolet-induced DNA lesions, its mechanistic role in damage-induced cohesion is unclear. Here, we show that transcriptional regulation is perturbed in the persistent absence of Polη. We find that Polη preferably associates with certain types of promoters, although its role in transcription might be indirect. By testing mutants that perturb histone exchange or regulation of transcription, as well as through inhibiting transcription, we show that transcriptional deregulation negatively affects formation of damage-induced cohesion. This supports the connection between transcriptional deregulation and deficient damage-induced cohesion in the Polη null mutant. Importantly, our study provides new insight into formation of damage-induced cohesion after replication, which will be interesting to explore further.

## Introduction

Dynamic disassembly and reassembly of nucleosomes—the building blocks of chromatin—facilitates processes such as replication and transcription. During the course of chromatin assembly, the canonical histones are exchanged with histone variants or post-translationally modified histones. This affects the physical and chemical properties of nucleosomes, as well as chromatin accessibility. Replication-independent nucleosome assembly, or so-called histone exchange, aids and regulates RNA polymerase II (RNAPII) passage through the nucleosomes during transcription initiation and elongation [1]. This is accomplished through histone chaperones, in concert with histone modifying enzymes and chromatin remodelers [2].

Transcription is not only the instrument for gene expression, but is also connected to cohesin localization on chromosomes. Cohesin is one of the structural maintenance of chromosomes (SMC) protein complexes, with the core formed by Smc1, Smc3 and the kleisin Scc1. Cohesin dynamically associates with chromosomes at intergenic regions of convergent genes, possibly as a result of active transcription [3,4]. Cohesin and its chromatin loader Scc2 have been implicated in gene regulation [5–7] and also in spatial organization of chromosomes into topologically associated domains (TADs) through DNA loop extrusion [8–12].

In addition to the roles described above, the canonical role of cohesin is to mediate sister chromatid cohesion. Cohesin is recruited to chromatin by the cohesin loading complex Scc2-Scc4 from late $G_1$ phase in *S. cerevisiae* [13], and continuously through the cell cycle [14,15]. During S-phase, cohesin becomes cohesive through acetylation of Smc3 by the acetyltransferase Eco1 [16–18]. The established sister chromatid cohesion is then maintained until anaphase [19], ensuring faithful chromosome segregation.

At the end of S phase, Eco1 is targeted for degradation. However, induction of double strand breaks (DSBs) post-replication ($G_2$/M) is sufficient to stabilize Eco1 [20,21]. Presence of active Eco1 then allows generation of damage-induced cohesion in $G_2$/M, which is established close to the break, and also genome wide on undamaged chromosomes [22–24]. We previously showed that Polymerase eta (Polη), one of the three translesion synthesis (TLS) polymerases in *S. cerevisiae*, is specifically required for genome wide damage-induced cohesion [25].

Polη (encoded by the *RAD30* gene) is well characterized for bypassing bulky lesions induced by ultraviolet irradiation [26], yet emerging evidence suggest that Polη also exhibits TLS-independent functions [27]. Polη is the only TLS polymerase required for damage-induced cohesion [25], independently of its polymerase activity, but dependent on Polη-S14 phosphorylation; potentially mediated by the cyclin dependent kinase, Cdc28 [28]. However, the underlying role of Polη in damage-induced cohesion remains unclear. Thus, absence of

Polη does not affect break-proximal damage-induced cohesion or DSB repair. Lack of Polη also does not perturb Eco1 stabilization, cohesin chromatin association or Smc3 acetylation after induction of DSBs in $G_2/M$ [25].

Based on the following two observations, we hypothesized that active transcription facilitates damage-induced cohesion genome wide. First, Polη is enriched at actively transcribed regions, and required for expression of several active genes in *S. cerevisiae* [29]. Second, activated transcription leads to establishment of local *de novo* cohesion in *S. pombe* [30]. In other words, it is possible that transcription is deregulated in the Polη null mutant, and that this subsequently affects formation of damage-induced cohesion. Here, we showed that chromatin association of RNAPII is reduced in the absence of Polη, or if Polη-S14-phosphorylation is abolished. In addition, the transcriptional program in the Polη null mutant (*rad30Δ*) is altered both before and after DSB induction, with expression of genes involved in chromatin assembly and positive transcription regulation being downregulated compared to WT cells. Perturbing histone exchange at promoter regions by a *HIR1* or *HTZ1* deletion negatively affects damage-induced cohesion, in a similar fashion as in *rad30Δ* cells, while deletion of the transcription elongation regulator *SET2* suppresses the lack of damage-induced cohesion in the *rad30Δ* mutant. Importantly, the potential linkage between transcription and formation of damage-induced cohesion was further supported by the fact that inhibiting transcription negatively affects its formation. Taken together, our results suggest that the transcription deregulation in the Polη null mutant is relevant to its deficient damage-induced cohesion. This provides new insight into formation of damage-induced cohesion post-replication, of importance for future investigations.

## Results

### Chromatin association of RNAPII is reduced in the Polη null and *Polη-S14A* mutants

To test if active transcription is correlated with generation of damage-induced cohesion, we initially assessed sensitivity of the damage-induced cohesion deficient *rad30Δ* and *Polη-S14A* cells to transcription elongation inhibitors. Viability of both mutants decreased when exposed to actinomycin D (Fig 1A). In addition, consistent with a previous report [29], *rad30Δ* cells were sensitive to mycophenolic acid (MPA). This was also true for the *Polη-S14A* point mutant (Fig 1A). Sensitivity of both mutants to MPA was reversed by supplementing the media with guanine (Fig 1A), verifying that it was due to depletion of the guanylic nucleotide pool [31].

Sensitivity to elongation inhibitors might be due to reduced transcriptional capacity. We therefore monitored chromatin association of Rpb1, the largest subunit of RNAPII, in these mutants. Binding of Rpb1 at promoters and coding regions of selected active genes was reduced in both *rad30Δ* and *Polη-S14A* mutants compared to WT cells (Fig 1B). The reduced chromatin association was accompanied by an increased level of total Rpb1 (Figs 1C and S1A). Furthermore, Rpb1 stability in the *rad30Δ* and *Polη-S14A* mutants was not affected, regardless of DSB induction (Figs 1C, 1D and S1A and S1B). Here and throughout the study the DSBs were induced at the *MAT* locus on chromosome III ($P_{GAL}$-*HO*) for one-hour, unless otherwise stated. These results together suggest that Polη may facilitate chromatin association of RNAPII for proper transcription initiation and elongation, likely through phosphorylation of Polη-S14 but independently of DNA damage.

### Transcription is perturbed in *rad30Δ* mutants

To further pinpoint a potential connection between transcription and formation of damage-induced cohesion, we focused on the *rad30Δ* mutant for the following investigations. To begin

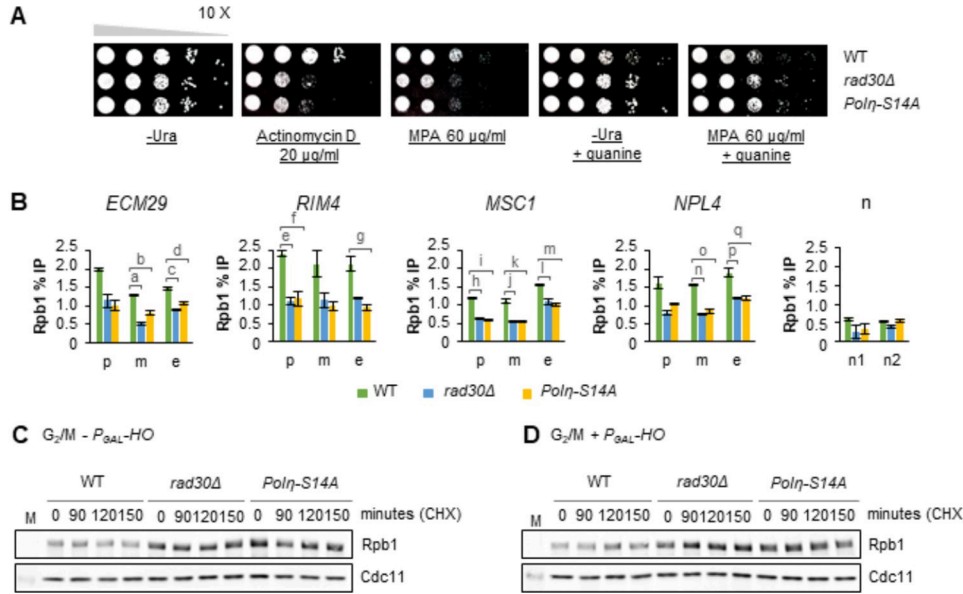

**Fig 1. Chromatin association of RNAPII is reduced in the Polη null and *Polη-S14A* mutants.** (A) Spot assay to monitor sensitivity of the *rad30Δ* and *Polη-S14A* mutants to the transcription elongation inhibitors, actinomycin D and mycophenolic acid (MPA). Tenfold serial dilutions of indicated mid-log phase cells on controls (-Ura plate ± guanine), and drug-containing plates, after 3 days incubation at room temperature. (B) ChIP-qPCR analyses to determine chromatin association of Rpb1 in indicated strains, on selected actively transcribed genes in $G_2/M$ arrested WT cells. Error bars indicate the mean ± STDEV of two independent experiments. Statistical differences compared to the WT cells at indicated position were evaluated by One-way ANOVA, Tukey post hoc test. The respective *p* values ($<0.05$) for each mutant relative to WT are (a) 0.000, (b) 0.004, (c) 0.010, (d) 0.026, (e) 0.039, (f) 0.044, (g) 0.034, (h) 0.000, (i) 0.000, (j) 0.010, (k) 0.011, (l) 0.026, (m) 0.017, (n) 0.003, (o) 0.004, (p) 0.047, (q) 0.047. p,promoter; m, mid; e, end of gene body. n1 and n2, low-binding controls. (C-D) Western blot analysis of Rpb1 stability. $G_2/M$ arrested cells from indicated strains, with or without one-hour $P_{GAL}$-HO break induction, were pelleted and resuspended in media containing cycloheximide (CHX) to monitor Rpb1 protein levels without further protein synthesis. Cdc11 was used as loading control. M, protein marker.

with, we analyzed gene expression of $G_2/M$ arrested WT and *rad30Δ* cells, before and after one-hour break induction, by RNA-sequencing analysis (RNA-seq). Prior to RNA-seq, $G_2/M$ arrest and break induction were confirmed (S2A and S2B Fig). Principal component analysis (PCA) showed that the individual data sets were distributed as distinct clusters (S2C Fig). Differences in gene expression patterns between WT and *rad30Δ* cells were readily observed before break induction, with 395 genes upregulated and 439 genes downregulated in the $G_2/M$ arrested *rad30Δ* mutant (Fig 2A). In response to DSB induction, the WT cells showed 473 genes up- and 519 genes down-regulated (Fig 2B), whereas there were 360 genes up- and 230 genes down-regulated in the *rad30Δ* mutant (Fig 2C and S1 Data). While the differentially expressed genes in WT and *rad30Δ* cells after break induction significantly overlapped (S2D Fig) and trended in the same direction, the up- and down-regulation after DSBs was of greater magnitude in the WT cells (Fig 2D and 2E). This implies that the response to break induction in the *rad30Δ* cells is similar, but relatively attenuated in comparison to the response in WT cells. Furthermore, we noted that short genes were preferentially upregulated compared to long genes in WT cells after DSB induction (Fig 2F), similar to the reported gene length dependent changes of expression after UV exposure [32,33]. In contrast, differential expression after DSBs is independent of gene length in the *rad30Δ* mutant (Fig 2F), further indicating a difference between WT and *rad30Δ* cells in their transcriptional responses. From these results we

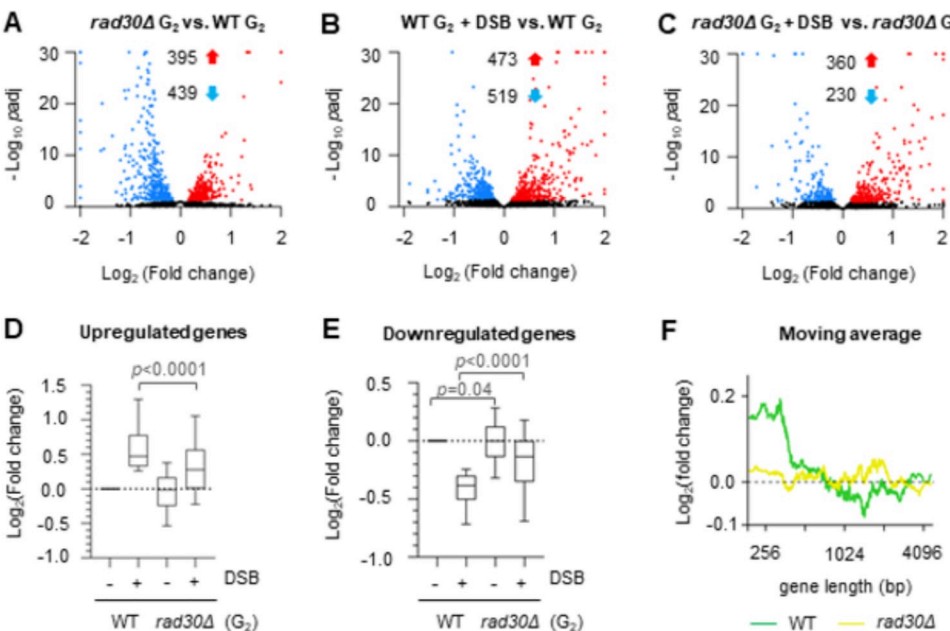

**Fig 2. Transcription is perturbed in *rad30Δ* mutants.** (A-C) Volcano plots showing differentially expressed genes between WT and *rad30Δ* cells, before and after DSBs, determined by RNA-seq. Each dot represents one gene. Red and blue dots represent up- and down-regulated genes respectively. Numbers of differentially expressed genes (*p*adj < 0.05) are indicated. Black dots indicate genes without significant changes in expression. *p*adj, adjusted *p* value. (D-E) Comparisons between expression level of genes significantly up (D) or downregulated (E) in the WT+DSB relative to the $G_2$/M arrested WT cells, and expression of the same set of genes in the *rad30Δ* mutant, based on RNA-seq analysis. Significant differences compared to the WT cells were evaluated by paired t-test. (F) Plot of fold change moving median, sorted by length (300 genes/window) to monitor the trend of gene expression after DSBs in relation to gene length, comparing WT and *rad30Δ* cells. Fold change values were based on the changes of gene expression in WT and *rad30Δ* cells after DSBs, determined by RNA-seq.

conclude that *RAD30* deletion leads to transcription deregulation, both in unperturbed $G_2$/M phase and in response to break induction.

## Polη is more frequently associated with closed-, FN- and TATA-containing promoters

As an attempt to better understand the possible role of Polη during transcription, we used published datasets to analyze if the deregulated genes in *rad30Δ* cells were associated with specific types of promoters, in a similar manner as reported [34]. These datasets classify genes according to type of promoter: (i) open/closed promoters, either with or without a nucleosome free region [35], (ii) promoters with fragile/stable nucleosome (FN/SN), defined by sensitivity of the -1 nucleosome to MNase digestion [36], and (iii) the canonical TATA-containing or TFIID dominated promoters [37,38]. Notably, a significant number of downregulated genes in $G_2$/M arrested *rad30Δ* cells were classified under the group of closed promoters (Table 1). In addition, the up- and down-regulated genes in $G_2$/M arrested *rad30Δ* cells were dominated by TATA-containing promoters (obs/exp>1). These data imply that Polη more frequently associates with promoters in closed configuration and TATA-containing promoters, primed for transcriptional activation in $G_2$/M phase. Interestingly, this prediction was supported by a Polη-ChIP-sequencing analysis. By monitoring genome-wide distribution of Polη during $G_2$/M phase, we found that Polη was enriched 100 bp upstream of transcription start sites (TSSs) and downstream of transcription end sites (TESs) but not at gene bodies (S3A Fig).

**Table 1. Association of differentially expressed genes with promoter type in G$_2$/M arrested *rad30Δ* cells.**

| *rad30Δ* G2 vs. WT G2 | upregulated (395) | | | | downregulated (439) | | | |
|---|---|---|---|---|---|---|---|---|
| | overlap | obs/exp | *p* values | | Overlap | obs/exp | *p* values | |
| closed promoter (1596) | 118 | 1.1 | 0.046 | | 146 | **1.3** | 3.309e-04 | * |
| open promoter (3504) | 228 | 1.0 | 0.459 | | 237 | 0.9 | 0.077 | |
| FN promoter (1953)[a] | 139 | 1.1 | 0.086 | | 156 | 1.1 | 0.054 | |
| SN promoter (3066)[b] | 206 | 1.0 | 0.223 | | 245 | 1.1 | 0.008 | |
| TATA-containing (1090) | 96 | **1.4** | 6.726e-04 | * | 132 | **1.7** | 5.069e-11 | * |
| TFIID-dominated (5130) | 299 | 0.9 | 7.636e-06 | * | 326 | 0.9 | 4.377e-08 | * |

Number of genes in each group is indicated in parentheses. The numbers in bold indicate that the overlap is higher than expected, observation/expectation (obs/exp)>1. Asterisks indicate significant overlap ($p<0.001$), evaluated as described in materials and methods. [a]FN: fragile nucleosome, [b]SN: stable nucleosome.

Furthermore, Polη more frequently associated with closed, FN and TATA-containing promoters (Fig 3A–3C); rather than the open, SN and TFIID-dominated promoters. To better understand if the transcriptional deregulation seen in *rad30Δ* cells was a direct or indirect effect, we set out to compare gene expression in the *rad30Δ* mutant with that of a 'Polη-degron' strain. The 'Polη-degron' strain harbors a combined auxin-inducible degron (AID) and Tet-off system [39,40], which allowed us to temporally deplete Polη during G$_2$/M by addition of auxin and doxycycline (S3B Fig). We initially selected six Polη-bound or unbound promoters according to the Polη-ChIP-sequencing analysis (Fig 3D), and tested if expression of the corresponding genes was affected in *rad30Δ* and Polη-depleted cells. Expression of the Polη-bound genes (*RIM4, PUT1, ECM29*) was as expected reduced in *rad30Δ* cells (Fig 3E), although Polη-binding at the *ECM29* promoter was less pronounced (Fig 3D). This was on the contrary not the case if depleting Polη specifically during G$_2$/M phase (Figs 3F and S3B). Since expression of the *DDR48* and *PUT1* genes was upregulated in Polη-depleted cells, as compared to the untreated control (Fig 3F), we examined expression of five additional Polη-associated genes (S3C and S3D Fig). Expression of these genes, however, showed no difference between auxin/doxycycline-treated and untreated cells (S3D Fig). Thus, Polη appears to play an indirect role during transcription, and the transcriptional deregulation observed in *rad30Δ* cells is likely accumulated through multiple cell cycles under persistent absence of Polη.

## Genes involved in chromatin assembly and positive transcription regulation pathways are downregulated in the absence of Polη

To gain mechanistic insight into the diverse transcriptional responses detected in WT and *rad30Δ* cells, differential gene expression between WT and *rad30Δ* cells (before and after DSBs) were analyzed by Gene Set Enrichment Analysis (GSEA), followed by generation of enriched pathway maps with Cytoscape as shown in Fig 4. The gene sets under each annotated group are listed in S2 and S3 Data. During G$_2$/M arrest, genes that belong to biological pathways such as chromatin assembly and positive transcription regulation were downregulated in *rad30Δ* compared to WT cells (Fig 4A). Consistent with downregulation of genes involved in the chromatin assembly pathway, we observed that the global nucleosome occupancy of *rad30Δ* cells was moderately increased compared to WT cells (S4A Fig). Although this may raise a concern about cohesin binding in *rad30Δ* cells, as nucleosome-free regions at promoters are required for cohesin loading [6,41], we previously noted that absence of Polη does not result in apparent differences in overall cohesin binding [25]. However, by revisiting our published Scc1 ChIP-sequencing dataset (GSE42655) and performing genome-wide meta-analysis, we found that association of cohesin around TSS was increased in *rad30Δ* compared to WT

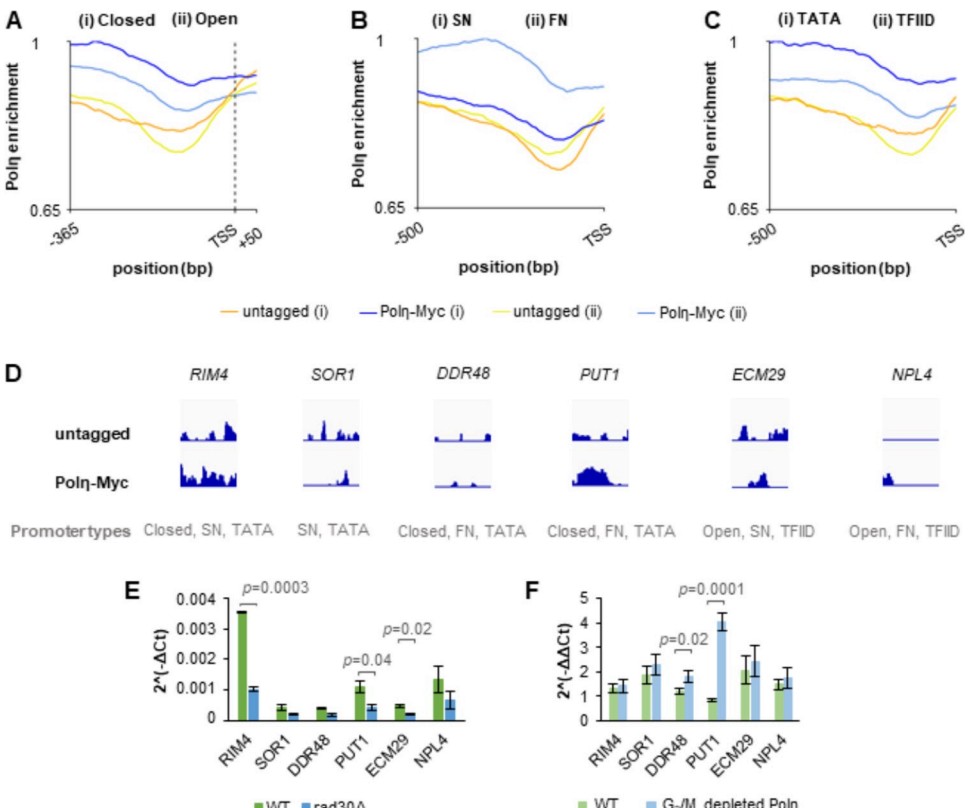

**Fig 3. Polη is more frequently associated with closed-, FN- and TATA-containing promoters.** (A) Metagenome plot showing accumulation of Polη at closed or open promoters, from 365 bp upstream to 50 bp downstream of the transcription start site (TSS) in $G_2$/M phase. The samples were first normalized to their respective input and then the values were scaled to the maximum value of the plot. (B-C) As in (A), except plotting accumulation of Polη at 500 bp upstream of TSS, to compare its relative enrichment at SN or FN promoter in (B); at TATA or TFIID-dominated promoters in (C). (D) Representative Integrative Genomics Viewer (IGV) tracks of Polη-ChIP-seq at selected promoters, with all Y-axes in the same scale. The samples were normalized to their respective input and the library size. (E) Expression of selected genes in $G_2$/M arrested WT and *rad30Δ* cells, measured by RT-qPCR. Error bars indicate the mean ± STDEV of two independent experiments. Statistical differences between WT and *rad30Δ* cells were evaluated by two-tailed t-test. (F) Expression of selected genes with or without depletion of Polη during $G_2$/M, measured by RT-qPCR. Polη was temporally depleted by addition of auxin and doxycycline; the mock control was denoted as 'WT'. The differences of ΔCt values between samples before and after addition of drugs were calculated as ΔΔCt, presented as $2^{-\Delta\Delta Ct}$ in the graph. The same calculations were applied to the mock control. Error bars indicate the mean ± STDEV of three independent experiments. Statistical differences between WT and Polη-depleted cells were evaluated by two tailed t-test.

cells (S4B and S4C Fig). Notably, this increased binding was not found around TES (S4D and S4E Fig), and was independent of DSB induction (S4B–S4E Fig). This could reflect that cohesin bound at TSS becomes less dynamic when transcription is dysregulated, as in *rad30Δ* cells.

When comparing gene expression after break induction, the pathways illustrated in Fig 4B were clearly differentially regulated between WT and *rad30Δ* cells. The nucleotide metabolism and amino acid metabolism pathways in WT cells, for instance, were upregulated to less extent compared to *rad30Δ* cells. This further indicates deregulation of gene expression in the *rad30Δ* mutant. Considering the fact that DNA damage response (DDR) proteins contribute to formation of damage-induced cohesion [22,24], we looked into the DDR pathway after DSB induction. Despite that some genes belonging to the cellular response to DNA damage stimulus pathway (GO: 6974) were upregulated in WT cells after DSB induction, this pathway was

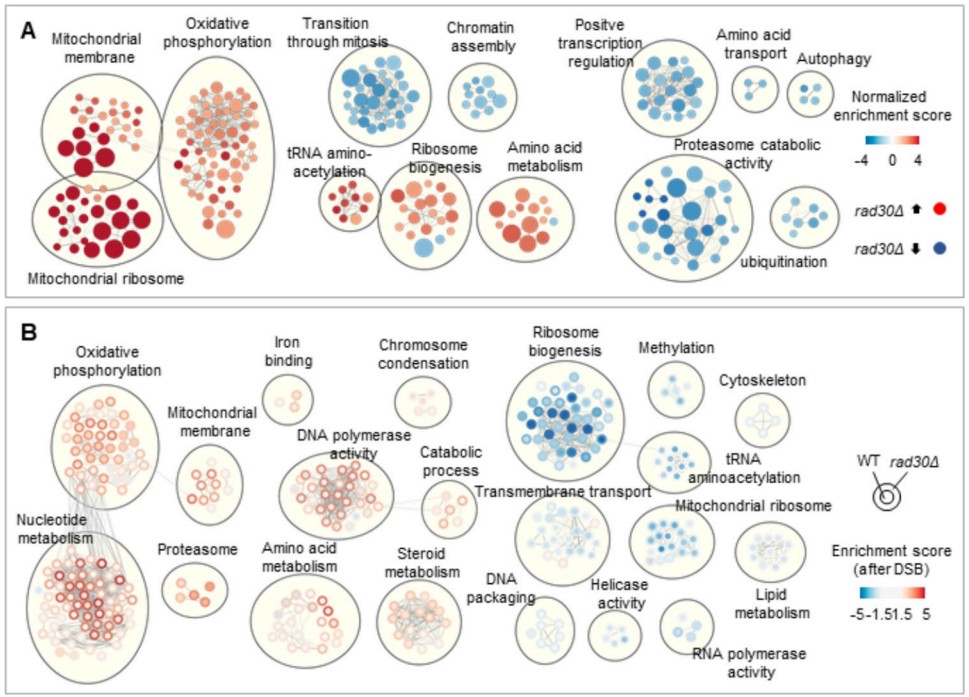

**Fig 4. Genes involved in chromatin assembly and positive transcription regulation pathways are downregulated in the absence of Polη.** (A) Relatively enriched pathways in $G_2$/M arrested WT and *rad30Δ* cells, plotted with Cytoscape after gene set enrichment analysis (GSEA). The GSEA was performed with gene lists ranked by $\log_{10} p$ value (multiplied by the sign of the fold change) of each gene. The number of genes in each gene set is proportional to the circle size. Lines connect gene sets with similarity greater than 0.7. All gene sets have FDR < 0.05. (B) Gene set enrichment analysis after DSB induction, plotted with Cytoscape to depict the difference between WT and *rad30Δ* cells in up- or down-regulation of indicated pathways after DSBs. Gene expression of WT and *rad30Δ* cells after DSBs was compared to that of respective $G_2$/M arrested cells. GSEA was performed as in (A). The lines indicate the same as in (A). All gene sets have FDR < 0.05 and a normalized enrichment score > 2 for at least one of the WT or *rad30Δ* cells.

overall not significantly enriched. In addition, activation of the DNA damage checkpoint, as indicated by phosphorylation of Rad53, was only observed during the recovery period after DSB induction in both WT and *rad30Δ* cells (S4F and S4G Fig), with no difference in cell cycle progression between populations (S4H Fig). These results indicate that the lack of damage-induced cohesion in *rad30Δ* cells is not due to a possible difference in activation of the DNA damage checkpoint. Furthermore, in response to DSBs, expression of the acetyltransferase *ECO1* was not enhanced in either WT or *rad30Δ* cells (S4I Fig). Altogether, this made it plausible to investigate the potential connection between transcription and damage-induced cohesion, and for this we focused on two of the upregulated gene sets in WT cells before DSB induction—chromatin assembly and positive transcription regulation.

## Deleting *HIR1* leads to partially deficient damage-induced cohesion

To assess if transcriptional activity is related to generation of damage-induced cohesion, we utilized a genetic approach by testing mutants which in theory should mimic or reverse the transcriptional deregulation in *rad30Δ* cells. One of the interesting candidates was Hir1 (a component of the HIR complex) that is known to be involved in chromatin assembly. The HIR complex and the histone chaperone Asf1 mediate histone H3 exchange with post-translationally modified H3, independently of replication [42,43]. The exchange mainly takes place at promoters and correlates with active transcription. However, basal H3 exchange also occurs to

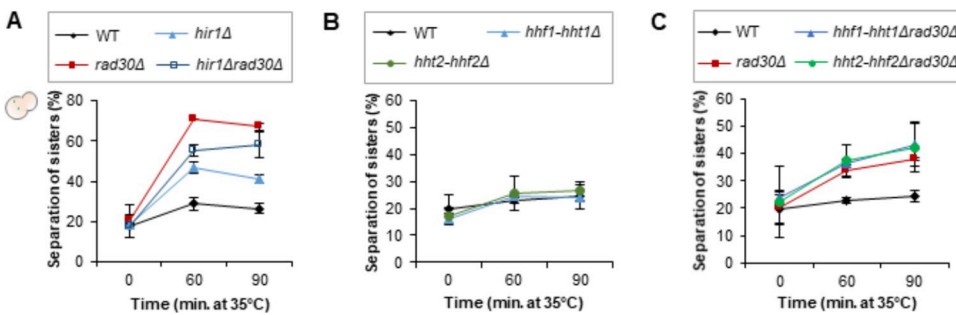

**Fig 5. Deleting _HIR1_ leads to partially deficient damage-induced cohesion.** (A) Damage-induced cohesion assays of the _hir1Δ_ single and _hir1Δrad30Δ_ double mutants after $P_{GAL}$-_HO_ induction, performed as illustrated in S5A Fig. Means ± STDEV from two independent experiments are shown. Two-hundred cells were counted for each time point, in each experiment. (B-C) Damage-induced cohesion assays of the _hhf1-hht1Δ_ and _hht2-hhf2Δ_ mutants after $P_{GAL}$-_HO_ induction, performed as in (A). Means ± STDEV from two independent experiments are shown. At least two-hundred cells were counted for each time point, in each experiment.

poise inactive promoters for optimal transcription [44,45]. Therefore, the relevance between transcriptional activation and formation of damage-induced cohesion could be investigated through the _hir1Δ_ mutant.

To monitor damage-induced cohesion, DSBs and ectopic $P_{GAL}$-_SMC1-MYC_ expression were induced by addition of galactose to $G_2/M$ arrested cells. Due to the _smc1-259 ts_ background, cohesion established during replication was inactivated by raising the temperature. Damage-induced cohesion generated with the ectopic Smc1-Myc was examined with an integrated TetO/TetR-GFP array on Chr. V (illustrated in S5A Fig). $G_2/M$ arrest, break induction and protein expression of the ectopic Smc1-Myc were confirmed for all experiments, with examples shown in S5B–S5D Fig. Interestingly, formation of damage-induced cohesion was partially deficient in the _hir1Δ_ mutant, while the _hir1Δrad30Δ_ double resembled the _rad30Δ_ single mutant, although with slower sister separation (Fig 5A). This indicated that Hir1 and Polη are both required for efficient damage-induced cohesion; possibly acting in the same pathway.

By using the _hir1Δ_ mutant, we determined if the HIR/Asf1-dependent histone exchange affected formation of damage-induced cohesion. However, the observed deficiency of the _hir1Δ_ cells might be due to de-repression of histone genes, as the HIR complex also negatively regulates expression of histone genes [46,47]. If so, reducing the histone gene dosage should be beneficial for the _rad30Δ_ mutant in generation of damage-induced cohesion. Yet, deletion of any H3-H4 coding gene pair (_HHT1-HHF1_ and _HHT2-HHF2_) did not affect formation of damage-induced cohesion, neither on their own nor in _rad30Δ_ cells (Fig 5B and 5C). This indicates that the partial deficiency of the _hir1Δ_ mutant is not due to altered histone gene dosage, and points to a need for histone exchange during transcription for formation of damage-induced cohesion.

## Perturbing histone exchange at promoters negatively affects formation of damage-induced cohesion

To further investigate the effect of pertubing histone exchange on formation of damage-induced cohesion, we tested _HTZ1_ deleted cells. Htz1, the histone variant of H2A, is preferentially incorporated at basal/repressed promoters. It is however susceptible to be evicted from the nucleosome, and that in turn promotes its exchange for H2A. This facilitates transcriptional activation [48,49], and relieves the +1 nucleosome barrier to RNAPII [50,51]. Since the

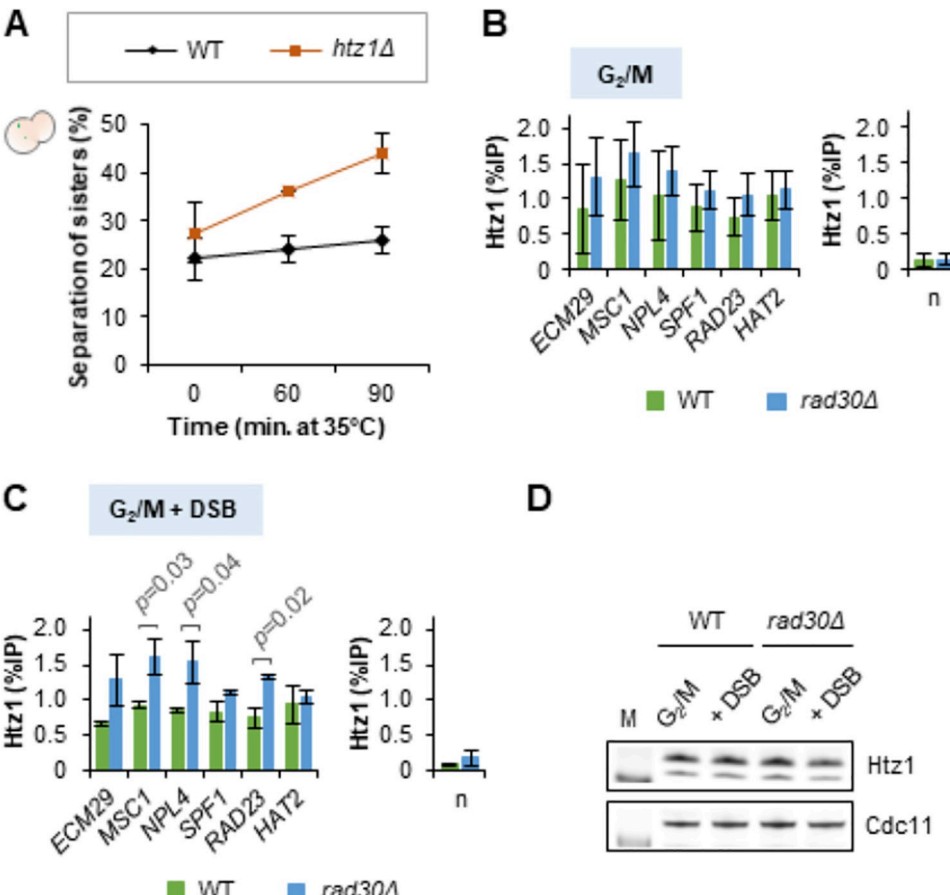

**Fig 6. Perturbing histone exchange at promoters negatively affects formation of damage-induced cohesion.** (A) Damage-induced cohesion assay of the *htz1Δ* mutant after γ-irradiation, performed according to the procedure described in the materials and methods. Means ± STDEV from two independent experiments are shown. For each experiment, at least two-hundred cells were counted for each time point. (B-C) ChIP-qPCR analyses to determine Htz1 occupancy at promoters of selected genes, before (B) and after DSB induction (C) in G$_2$/M arrested WT and *rad30Δ* cells. *SPF1*, *RAD23* and *HAT2* are located at the left arm of chromosome V, where damage-induced cohesion was monitored. Error bars indicate the mean ± STDEV of three independent experiments for (B) and two independent experiments for (C). Statistical differences compared to the WT cells were evaluated by t-test. n, low-binding control. (D) Western blot analysis of the total Htz1 protein level in WT and *rad30Δ* cells, before and after DSB induction during G$_2$/M phase. Cdc11 was used as loading control. M, protein marker.

*htz1Δ* mutant does not respond to P$_{GAL}$-*HO* induction [52], γ-irradiation was utilized as source of DSB induction (see materials and methods). Similar to the *hir1Δ* mutant (S6A and S6B Fig), the *htz1Δ* mutant showed impaired damage-induced cohesion (Fig 6A). In contrast to a previous report [53], we did not observe a cohesion maintenance defect due to *HTZ1* deletion (S6C Fig).

Since Htz1 is required for formation of damage-induced cohesion, we investigated if there was a difference in Htz1 occupancy at promoters between WT and *rad30Δ* cells. For this, we focused on three of the genes analyzed for Rpb1 binding in Fig 1B and three additional genes around the *URA3* on Chr. V, where we monitored damage-induced cohesion. We selected genes with TATA-less promoters for analyses because Htz1 is relatively enriched at these promoters [48,49]. Interestingly, Htz1 occupancy at some of the selected promoters was increased in *rad30Δ* compared to WT cells, particularly after DSB induction in G$_2$/M (Fig 6B and 6C). Despite this difference, the total protein level of Htz1 was similar between WT and *rad30Δ*

cells (Fig 6D). This indicates that the Htz1/H2A exchange at certain promoters was reduced in the absence of Polη, especially in response to DSBs. These results were in line with *hir1Δ* and *htz1Δ* cells being deficient in damage-induced cohesion (Figs 5A, 6A and S6B), and suggest that perturbing histone exchange at promoters negatively affects formation of damage-induced cohesion.

### Transcriptional deregulation leads to deficient damage-induced cohesion

In addition to the *hir1Δ* and *htz1Δ* mutants, we used a *set2Δ* mutant to test if transcriptional regulation is correlated with generation of damage-induced cohesion. Set2 mediates co-transcriptional H3K36 methylation (H3K36me1/2/3). This promotes restoration of chromatin to the pretranscribed hypoacetylation state and represses histone exchange at coding regions after transcription elongation [54–56]. Presence of Set2 at promoters also suppresses transcription initiation of certain basal repressed genes [57–59]. Interestingly, a *set2Δ* mutant was reported to suppress sensitivity of certain transcriptional elongation factor mutants to 6-azauracil [59], a mechanistic analog of MPA [60,61]. As we showed that *rad30Δ* cells are sensitive to transcription elongation inhibitors (Fig 1A), we tested if deletion of *SET2* would also rescue *rad30Δ* cells from this sensitivity. The *set2Δ* mutant was insensitivite to MPA or actinomycin D, and masked sensitivity of *rad30Δ* cells especially to actinomycin D (Fig 7A). Through this genetic interaction, we then tested if deletion of *SET2* would also suppress the deficiency of *rad30Δ* cells in damage-induced cohesion. While the *set2Δ* mutant resembled the WT cells in formation of damage-induced cohesion, deletion of *SET2* remarkably suppressed the lack of damage-induced cohesion in the *rad30Δ* mutant (Fig 7B). In addition, since removing *SET2* has been shown to cause an increased RNAPII association towards the 3'-end of actively transcribed genes [62], we monitored chromatin association of Rpb1 in the *set2Δrad30Δ* mutant. Absence of Set2 in $G_2$/M arrested *rad30Δ* cells to some extent compensated for the reduced Rpb1 binding in *rad30Δ* cells (Fig 7C–7E). This trend was however not observed after DSB induction (S7A–S7C Fig). Considering that the differentially expressed genes in WT and *rad30Δ* cells after DSB induction significantly overlapped (S2D Fig), these data together suggest that general transcription regulation during $G_2$/M phase influences formation of damage-induced cohesion.

To further examine this idea, we inhibited transcription and monitored formation of damage-induced cohesion after γ-irradiation. Transcription inhibition was achieved through the anchor-away system, which uses rapamycin to induce heterodimerization of the anchor (Rpl13A-FKBP12) and the FRB-tagged target, in our case Rpb1 [63], thereby excluding Rpb1 from the nucleus (S7D Fig). To avoid toxicity of rapamycin and its effect on transcription, the rapamycin binding protein Fpr1 and the rapamycin target Tor1 were either deleted or mutated in our 'Rpb1-anchor away' strain ([63–65] and S1 Table). Anchoring Rpb1 away from the nucleus by 1-hour rapamycin treatment caused approximately two-fold reduction in expression of selected genes (S7E Fig), without triggering early DNA damage response (S7F Fig), or compromising the protein level of the $P_{GAL}$-driven ectopic Smc1-Myc (S7G Fig). In line with previous experiments, without addition of rapamycin, damage-induced cohesion was formed after exposure to γ-irradiation (Fig 7F). However, transcription inhibition induced by rapamycin indeed negatively affected formation of damage-induced cohesion. Altogether, this supports the idea that transcription deregulation, as a consequence of persistent absence of Polη, is connected to deficienct damage-induced cohesion.

## Discussion

We previously showed that Polη is specifically required for genome wide damage-induced cohesion [25] but its mechanistic role in this process was unclear. The present study was

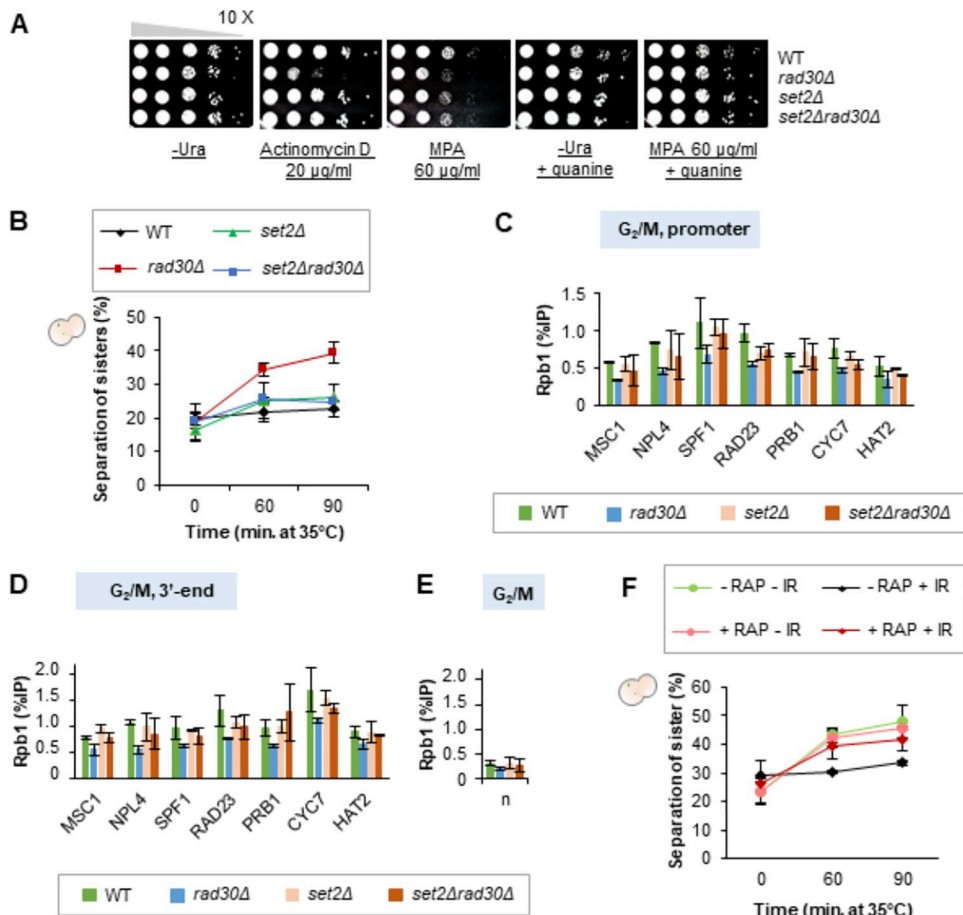

**Fig 7. Transcriptional deregulation leads to deficient damage-induced cohesion.** (A) Spot assay to monitor the effect of *SET2* deletion on the *rad30Δ* mutant sensitivity to the transcription elongation inhibitors, actinomycin D and mycophenolic acid (MPA). Tenfold serial dilutions of indicated mid-log phase cells on control (-Ura plate ± guanine) and drug-containing plates, after 3 days incubation. (B) Damage-induced cohesion assay of the *set2Δ* mutant after $P_{GAL}$-*HO* induction, performed as depicted in S5A Fig. Means ± STDEV from two independent experiments are shown. At least two-hundred cells were counted for each time point, in each experiment. (C-E) ChIP-qPCR analyses to determine chromatin association of Rpb1 at promoters and 3'-ends of selected genes, in indicated $G_2$/M arrested cells. Except *MSC1* and *NPL4*, the rest of the selected genes are located at the left arm of chromosome V, where damage-induced cohesion was monitored. Error bars indicate the mean ± STDEV of two independent experiments. n, low-binding control (n2 in Fig 1B). (F) Damage-induced cohesion assay of the Rpb1-anchor away strain. Gamma-irradiation (IR) was used as the source of DSBs. The assay was performed according to the procedure described in materials and methods. Rapamycin (RAP) was added to deplete Rpb1 from the nucleus. Means ± STDEV from three independent experiments are shown. At least two-hundred cells were counted for each time point, in each experiment.

initiated by the observation that Polη-deficient cells displayed altered transcriptional regulation, both in unchallenged $G_2$/M arrested cells and in response to DSBs. Transcription elongation deficiency was corroborated by increased sensitivity of Polη-deficient cells to transcription elongation inhibitors (Fig 1A). It could be argued that the sensitivity to actinomycin D would be a consequence of DNA damage because actinomycin D also inhibits topoisomerases [66], leading to formation of DSBs. However, since the *rad30Δ* mutant is insensitive to specific topoisomerase inhibitors, such as camptothecin and etoposide [67,68], this was less likely.

To know which pathways were affected in the absence of Polη, gene set enrichment analysis was performed after RNA-seq. We found that mitochondrial related pathways were enhanced

in *rad30Δ* cells, in contrast to downregulation of genes belonging to the chromatin assembly pathway (Fig 4A and 4B and S2 and S3 Data). This is an interesting observation since genes involved in the tricarboxylic acid cycle and oxidative phosphorylation pathways, which are related to mitochondria, were similarly upregulated in mutants with defective chromatin assembly [69].

To test if the lack of damage-induced cohesion in *rad30Δ* cells would be due to transcriptional dysregulation, we first tested the requirement of HIR/Asf1 mediated histone exchange for damage-induced cohesion, from the perspective of chromatin assembly. By deleting the *HIR1* gene, which is sufficient to disrupt the HIR/Asf1 interaction [43], we found that the *hir1Δ* mutant is partially deficient in damage-induced cohesion (Figs 5A and S6B). The role of the HIR complex in damage-induced cohesion might appear difficult to pinpoint since it is involved in multiple processes. We thus addressed the possible effect of HIR-dependent repression of histone genes [46] on formation of damage-induced cohesion. This possibility was however excluded because no effect of deleting H3-H4 gene pairs (Fig 5B and 5C) was observed in *rad30Δ* cells. The HIR complex has also been implicated in formation of a functional kinetochore [70] and heterochromatic gene silencing [71]. However, the chromatin assembly complex-1 (CAF-1) is redundant with the HIR complex in these processes. Deletion of Hir1 is thereby not likely to perturb other processes than histone exchange. We therefore suggest a direct role for HIR-dependent histone exchange in damage-induced cohesion.

Functional importance of Polη in transcription was proposed to depend on its polymerase activity [29], while its role in damage-induced cohesion was not [25]. The finding that transcription supports formation of damage-induced cohesion could therefore be seen as conflicting with the polymerase-independent role of Polη. However, we previously showed that the putative Polη-S14 phosphorylation is required for damage-induced cohesion, but not for cell survival after UV irradiation [28], which depends on Polη polymerase activity. In addition, the *Polη-S14A* mutant exhibits similar elongation inhibitor sensitivity and altered Rpb1 behaviour as the *rad30Δ* mutant (Fig 1A–1D). This together indicates that the polymerase activity is not the sole requirement for Polη in transcription.

To gain further insight into the role of Polη in transcription, we analyzed the types of promoters that Polη associates with (Table 1 and Fig 3). We found that the differentially expressed genes in $G_2$/M arrested *rad30Δ* cells, especially the downregulated genes, were relatively enriched for closed and TATA-containing promoters. In line with this, we showed by ChIP-sequencing that Polη preferentially occupies these two types of promoters. The closed promoters that lack a nucleosome free region, are known to regulate stress related genes [72]. This is consistent with the downregulation of stress response (GO:0033554) in $G_2$/M arrested *rad30Δ* cells (S2 Data, Ungrouped). Similarly, TATA-box containing genes are highly regulated and associated with stress response [37]. This together suggests that Polη could support transcription for proper stress response. Besides of closed and TATA-containing promoters, Polη was also found to be relatively enriched at FN promoters in our Polη-ChIP-sequencing analysis. The FN promoters typically regulate highly expressed or growth related genes [36]. However, the differentially expressed genes in *rad30Δ* cells did not significantly overlap with genes regulated by this type of promoter (Table 1), showing that their expression was not affected by persistent absence of Polη. Nevertheless, knowing the types of promoters that Polη preferentially associates with should be helpful for identification of its potential interactors during transcription. It would also be interesting to know how these preferences are correlated with formation of damage-induced cohesion in the future.

To further investigate if Polη acts directly or indirectly in transcription, we depleted Polη temporally during $G_2$/M and analyzed expression of selected genes. It turned out that in contrast to *rad30Δ* cells, expression of most tested genes was not affected (Figs 3E, 3F and S3D).

This suggests an indirect role of Polη in transcription, which would only perturb the transcriptional process if it is persistently absent from cells. We speculate that presence of Polη at specific promoters and Polη-S14-phosphorylation contribute to formation of a certain chromatin state, which is primed for proper transcriptional regulation. This is indicated by the reduced chromatin binding of Rpb1 in the Polη null and *Polη-S14A* mutants (Fig 1B), the preference of Polη for certain promoters (Fig 3A–3C) and the reduced Htz1/H2A exchange in *rad30Δ* cells (Fig 6C). In contrast, a temporal depletion of Polη during G$_2$/M would likely have less impact on chromatin state. Precisely how the persistent absence of Polη indirectly affects transcription remains to be investigated further.

Through perturbing histone exchange, removing a transcription elongation regulator (illustrated in S8 Fig), and inhibiting transcription by anchoring Rpb1 away from the nucleus (Figs 7F and S7D–S7G), we show that a regulated transcriptional response connected to chromatin assembly, potentially facilitates generation of damage-induced cohesion post-replication. Since establishment of sister chromatid cohesion is proposed to occur simultaneously with replication fork progression [14,73], in concert with replication-coupled nucleosome assembly [74], it is also possible that replication-independent nucleosome assembly (histone exchange) is utilized as an alternative platform for generation of damage-induced cohesion after replication (S8 Fig, WT). In support of this, deregulated transcription and reduced Htz1/H2A exchange in *rad30Δ* cells negatively affected formation of damage-induced cohesion (S8 Fig, *rad30Δ*).

Despite the subtle defect in chromosome segregation observed in the *rad30Δ* mutant [25], the importance of genome wide damage-induced cohesion remains to be determined. It might be relevant to the increased chromosome mobility in response to DSBs, which presumably facilitates the search of sequence homology for recombination [75,76]. Interestingly, chromosome mobility is at the same time constrained by sister chromatid cohesion [77]. Since unbroken chromosomes are known to be less mobile than broken chromosomes [75,76], formation of genome-wide damage-induced cohesion might further limit the movements of undamaged chromosomes, to reduce the chance of unfavorable recombinations.

In summary, we show that transcriptional deregulation driven by persistent absence of Polη leads to deficient damage-induced cohesion. Through a genetic approach, our study provides new insight into a potential linkage between histone exchange and generation of damage-induced cohesion post-replication. Further studies would be needed to understand how chromatin dynamics during transcription facilitate formation of genome wide damage-induced cohesion, and if damage-induced cohesion could restrict movements of undamaged chromosomes.

## Materials and methods

### Yeast strains and media

All *S. cerevisiae* yeast strains, listed in S1 Table, were W303 derivatives (*ade2-1 trp1-1 can1-100 leu2-3, 112 his3-11, 15 ura3-1 RAD5 GAL psi*+). To create null mutants, the gene of interest was replaced with an antibiotic resistance marker through lithium acetate based transformation. Some strains were crossed to obtain desired genotypes. Yeast extract peptone (YEP) supplemented with 40 μg/ml adenine was used as yeast media, unless otherwise stated.

### Spot assay

Cell culturing and subsequent serial dilutions were performed as described [28]. Each dilution was sequentially spotted on uracil drop-out (-Ura) media, containing actinomycin D, MPA, or solvent only (final 1.2% ethanol in plates). Guanine was supplemented at 0.3 mM final

concentration [78]. The plates were kept at room temperature and documented on the third day. Each spot assay was done at least twice.

## Protein extraction and western blotting

Whole cell extracts (WCEs) were prepared with glass bead disruption, TCA or a sodium hydroxide based method [79]. To monitor Rpb1 stability, cycloheximide (Sigma) was supplemented in media (final 100 μg/ml), and the protein extracts were prepared with sodium hydroxide based method. Bolt 4–12% Bis-Tris or NuPAGE 3–8% Tris-Acetate gels (Invitrogen) were used for electrophoresis, with Bolt MOPS, Bolt MES or NuPAGE Tris-Acetate SDS running buffer (Invitrogen). Proteins were transferred to nitrocellulose membranes with the Trans-blot Turbo system (Bio-Rad) or the XCell II Blot Module (Invitrogen). Antibody information is listed in the S2 Table. Odyssey Infrared Imaging and BioRad chemiluminescence system were used for antibodies detections. Image Studio Lite software was used for quantitation of protein bands.

## Chromatin immunoprecipitation (ChIP)-qPCR for Rpb1 and Htz1

ChIP was in essence performed as described with some modifications [25]. Cells were crosslinked with final 1% formaldehyde for 20 minutes at room temperature, followed by addition of final 125 mM glycine for 5 minutes. The cells were washed three times in 1X cold TBS and mechanically lysed using a 6870 freezer/mill (SPEX, CertiPrep). WCEs were subjected to chromatin shearing by sonication (Bandelin, Sonopuls) for chromatin fragments of 3–500 bp. Anti-Rpb1 and anti-Htz1 antibodies were coupled to protein A and protein G Dynabeads (Invitrogen) respectively for immunoprecipitation at 4˚C, overnight. Crosslinking of eluted IP and input samples was reversed, and DNA was purified. DNA analysis was performed by real time qPCR (RT-qPCR) using SYBR Green (Applied Biosystems), according to manufacturer's guidelines on an ABI Prism 7000 sequence detection system (Applied Biosystems). The genes of interest were selected based on the RNA-seq results. Primers used are listed in S3 Table. Each gene was analyzed with three technical repeats for each individual experiment. Statistical analysis was performed with SPSS statistics software (IBM).

## Polη-Myc ChIP

Preparation of WCEs for ChIP of Myc-tagged Polη was performed as described above, and the ChIP as in [29] with the following modifications. Sonicated cell lysates from 70–80 OD units were incubated with anti-MYC, rotating at 4˚C overnight. Protein G Dynabeads (Invitrogen) were then added for immunoprecipitation for 3.5 hours at 4˚C. After reversing cross-linking at 65˚C for 15 hours, the samples were treated with RNAse (50 μg/ml, final concentration) for 1 hour at 37˚C, and finally the chromatin was purified (PCR purification kit, Qiagene). DNA analysis was performed by ChIP Sequencing (see below).

## Total RNA extraction and RT-qPCR

For RNA-seq, $G_2/M$ arrested cells (about 9 $OD_{600}$) were harvested before and after 1-hour $P_{GAL}$-$HO$ break induction. Equal amount of samples were additionally collected at each timepoint as genomic DNA (gDNA) controls. The gDNA content of each sample was determined prior to total RNA extraction. Total RNA extracts were prepared with PureLink RNA Mini Kit (Invitrogen), with some modifications of the manufacture's guidelines. Collected cell pellets were washed once with SE mix (final 1 M sorbitol and 50 mM EDTA), and resuspended with 100 μl zymolyase lysis buffer (SE mix supplemented with final 3 mg/ml 100T zymolyase

(Sunrise Science) and 2.5 μl Ribolock (Invitrogen). The suspension was incubated at 30˚C for 60 minutes, followed by addition of 200 μl kit-provided RNA lysing buffer, supplemented with Ribolock. The rest of the procedure was performed according to the manufacture's guidelines. To elute total RNA from columns, the volume of RNase free water for elution was adjusted according to gDNA content of each sample. For each strain, equal volume of the total RNA extract was further purified with DNA-free Kit (Invitrogen).

For RT-qPCR, purified total RNA (300 or 650 ng) was spiked in with 1 ng luciferase control RNA (Promega) prior to cDNA synthesis. Luciferase was then used as the reference gene for data analyses [80], unless otherwise stated. Primers used are listed in S4 Table. Each gene was analyzed with three technical repeats for each individual experiment.

### RNA-sequencing and ChIP-sequencing data analyses

Total RNA samples prepared for RNA-seq (triplicates) were subsequently handled by Novogene for mRNA enrichment, library construction (250–300 bp insert cDNA library) and RNA sequencing (Illumina HiSeq X Ten, paired-end, 10 M reads). Quality controls were included for the total RNA samples and during the procedures for RNA-sequencing.

FASTQC (https://www.bioinformatics.babraham.ac.uk/projects/fastqc/) was used for quality control of the .fastq-files for both RNA- and ChIP-seq. Adapter and poor quality read trimming was performed with cutadapt [81]. The RNA-seq data was mapped with the splice-aware aligner HISAT2 [82]. The Scc1-ChIP-seq data was mapped using Bowtie [83] with the color-space option enabled, while the Polη-ChIP-seq data was mapped using Bowtie2 [84]. Afterwards the mapped files were sorted using samtools [85]. All three sets of sequencing data were aligned to the yeast genome version SacCer3 downloaded from UCSC genome browser. Duplicates in the mapped.bam-files were removed using MarkDuplicates (http://broadinstitute.github.io/picard) from the Picard toolset.

For the RNA-seq data set, the reads were counted per gene using featureCounts [86]. The count-files were imported into R and further analyzed using edgeR [87,88] for FPKM calculations and DESeq2 [89] for differential expression analysis. Differential expression analysis yielded fold-changes alongside significance for genes, additionally DESeq2 was used to generate principal component analysis plots. Genes with a total read count below 10 across all samples as well as those producing NAs (not available) in any of the comparisons for fold-change calculation were excluded from the analysis. As all four conditions showed a similar within-group variability in the PCA plot, for all fold-change calculations all samples were run together as opposed to subsetting the samples of interest e.g. WT G2 + DSBs vs. WT G2. This allowed for more accurate estimation of the dispersion parameter and in turn calculation of significance for the fold-changes. Also, the moving average of the fold-change was calculated by ordering the genes included in the DESeq2 dataset by length and then calculating the median of a window of 300 genes around these gene. No moving average was calculated for the 75 longest and shortest genes as they did not have an even number of genes on either site for moving average calculation.

For the Scc1-ChIP-seq dataset, cohesin peaks were called using MACS2 [90]. The files generated were then imported into R, where they were annotated using the package ChIPpeakAnno [91] with gene lists downloaded using the biomaRt package [92]. The lists of genes overlapping or with their gene end closest to the peak middle with cohesin peaks were read into ngs.plot [93] for metagenome analysis. After analysis had been performed, the data were replotted using the internal R plotting. Ngs.plot was also used to perform metagenome analysis for the Polη-ChIP-seq dataset at different promoter types. Additionally, the bigCompare command of the deepTools suite was used to generate bigWig files of the Polη-IPs normalized to

both their respective inputs and the library size [94]. These bigwig files were then loaded into the Integrative Genomics Viewer (IGV) for visualization [95,96].

Gene set enrichment analysis (GSEA) was performed using the Broad Institute software (http://www.broad.mit.edu/gsea) [97] using *S. cerevisiae* gene sets from the Xijin Ge lab (http://ge-lab.org/#/data) [98]. The GSEA enrichment map was created using the Enrichment-Map plugin [99] for Cytoscape [100], broadly following a published protocol [101]. Groupings were facilitated by the Cytoscape AutoAnnotate plugin [102]. In the comparison of WT vs. *rad30Δ* cells, only gene sets enriched with an adjusted *p*-value of < 0.05 were plotted. In the comparison of both WT and *rad30Δ* cells ± DSB induction, only gene sets enriched with an adjusted *p*-value of < 0.05 and a normalized enrichment score (NES) > 2 for either strain were plotted.

Statistical significance of the overlapping genes in the Venn diagrams and Table 1 were calculated using either a normal approximation or the hypergeometric probability formula. The online tool on http://nemates.org/MA/progs/overlap_stats.html was used for evaluation.

## Damage-induced cohesion assay and controls

All strains used harbor the *smc1* temperature sensitive allele (*smc1-259*). The experiments with the P$_{GAL}$-*HO* allele for DSB induction were performed as described [28], and illustrated in S5A Fig. The assay utilizing γ-irradiation as DSB source is described in S6A Fig. Considering that the *htz1Δ* mutant is benomyl sensitive [103], the strains used in this assay contain the P$_{MET}$–*CDC20* and *smc1-259 ts* alleles. The strains were grown in methionine drop-out media (-Met) to log phase at 23˚C. To arrest cells in G$_2$/M phase, expression of *CDC20* was repressed by replacing the media to YEP supplemented with Met (final 2 mM) and 0.1% glucose. Galactose (final 2%) was then added for 1.5 hours to induce expression of ectopic Smc1-Myc, driven by the *GAL* promoter. The cultures were subsequently split into half and resuspended in 1X PBS. One half for γ-irradiation (250 Gy), and another half as non-irradiated control. After 1-hour recovery in YEP media supplemented with galactose and Met, the temperature was raised to 35˚C and damage-induced cohesion was monitored for 90 minutes.

For the Rpb1-anchor away strain, damage-induced cohesion assay was performed as illustrated in S6A Fig with the following modifications. The culture was split after 1-hour *GAL*-induction, half for addition of rapamycin (final 1 µg/ml) and half for addition of DMSO as control. After 1-hour ± rapamycin treatment, the cultures were spun down and resuspended in PBS supplemented with benomyl (PBS/B). The following procedures were as depicted in S6A Fig, except the cells were allowed to recover for 30 minutes in YEP media supplemented with glucose and benomyl after ± γ-irradiation. Noted that after resuspension in PBS/B, the cultures were always supplemented with rapamycin or DMSO when changing media.

Proper G$_2$/M arrest, expression of the ectopic Smc1-Myc and DSBs induction in these assays were confirmed with FACS analysis, western blot, and pulsed-field gel electrophoresis (PFGE) respectively. Efficiency of γ-irradiation was analyzed with Southern blot after PFGE, with a probe for chromosome XVI, as described [104]. Rpb1-*in situ* staining was performed as described [28], using a specific anti-Rpb1 antibody.

## MNase digestion assay

G$_2$/M arrested cells were crosslinked *in vivo* with formaldehyde (final 0.5%), for 20 minutes at 23˚C. To quench the reaction, glycine (final 125 mM) was added in cultures for 10 minutes. The cells were then harvested and stored at -80˚C. Prior to MNase digestion, the cells were resuspended in pre-incubation solution (final 20 mM citric acid, 20 mM Na$_2$HPO$_4$, 40 mM EDTA, pH 8.0), with aliquots taken for cell-counting. The final volume of resuspension was

subsequently adjusted to have $4.5 \times 10^7$ cells/ml. The cells were pre-treated with freshly added 2-mercaptoethanol (2-ME, final 30 mM in pre-incubation buffer) for 10 minutes at 30˚C, followed by zymolyase treatment in zymolyase buffer (final 1 M sorbitol, 50 mM Tris-HCl (pH 7.5), 10 mM 2-ME and 1 mg/ml 100T zymolyase) for 30–35 minutes [105]. Converted spheroplasts were washed once with cold zymolyase buffer without 2-ME, resuspended in nystatin buffer (final 50 mM NaCl, 1.5 mM CaCl₂, 20 mM Tris-HCl (pH 8.0), 1 M sorbitol, and 100 μg/ml nystatin (Sigma), and then kept on ice temporarily.

The following MNase digestion was performed for each strain individually. Resuspended spheroplasts were sequentially added into the MNase aliquots (ranged from final 0.0125 to 0.1 U/ml, prepared in nystatin buffer), and incubated at 25˚C for 15 minutes. Reactions were stopped by adding 1% SDS/12 mM EDTA (final concentration) [106,107]. Subsequently, the spheroplasts were treated with RNase (final 0.02 μg/μl) at 37˚C for 45 minutes, followed by proteinase K (final 0.4 μg/μl) at 65˚C, overnight. The DNA samples were purified with phenol/chloroform extraction, precipitated with ethanol overnight and then resuspended in 1X TE. The samples (2.5 μg) were analyzed with gel electrophoresis (1.2% TAE agarose gel, at 35 V overnight) [105].

## Supporting information

**S1 Fig. Quantitation of Rpb1 levels.** (A-B) Relative amounts of Rpb1 after addition of water (A, control) or galactose (B) to induce $P_{GAL}$-*HO* DSB induction for one-hour, followed by cycloheximide (CHX) chase up to 150 minutes. Western blots from two independent experiments were quantified to compare Rpb1 levels (relative to Cdc11) between the indicated strains.
(TIFF)

**S2 Fig. Control experiments for RNA-seq and Venn diagrams of differentially expressed genes in indicated strains after DSB induction.** (A) FACS analysis to confirm benomyl-induced $G_2$/M arrest. 1G, 1-hour *GAL*-induction ($P_{GAL}$-*HO*). (B) PFGE analysis to monitor DSB induction on chromosome III. $G_2$, G2/M arrest; 1G as in (A). (C) PCA demonstrating distribution of independent data sets between groups and clustering of data sets within groups. (D) Venn diagrams showing overlaps of differentially expressed genes in WT and *rad30Δ* cells after DSBs, based on RNA-seq. The red and blue arrows indicate up- and down-regulated genes respectively. Statistical significance of the overlapping genes was evaluated as described in Materials and Methods, with * $p < 0.001$.
(TIFF)

**S3 Fig. Genome-wide distribution of Polη and additional gene expression analyses for Polη-depleted cells during $G_2$/M.** (A) Metagenome plot showing distribution of Polη, with 100 bp flanking regions upstream and downstream of the gene bodies during $G_2$/M phase. The samples were first normalized to their respective input and then the values were scaled to the maximum value of the plot. (B) Western blot to check depletion of Polη in $G_2$/M arrested cells. Final concentrations of auxin and doxycycline were 6 mM and 20 μg/ml respectively. IAA, auxin; dox, doxycycline; $t_0$, the 0-time point after addition of IAA/dox; $t_{1.5}$, 90 minutes after treatment. The drug solvents (50% ethanol and water) were added in the '-IAA/dox' mock control. The western blot image, including the protein marker, was cropped to show selected samples. Cdc11 was used as loading control. (C) Representative Integrative Genomics Viewer (IGV) tracks showing the differences in distribution of Polη at selected promoters. The samples were normalized to their respective input and library size. (D) Expression of selected genes with or without depletion of Polη during $G_2$/M, measured by RT-qPCR. Calculations

were the same as described in the legend of Fig 3F. Error bars indicate the mean ± STDEV of three independent experiments.
(TIFF)

**S4 Fig. The *rad30Δ* mutant showed increased nucleosome occupancy, but no difference in activation of DNA damage checkpoint and *ECO1* gene expression compared to WT cells.**
(A) Monitoring nucleosome occupancy based on sensitivity of cells to MNase digestions. The concentrations of MNase were 0, 0.0125, 0.025, 0.05, 0.1 U/ml (final). One representative gel electrophoresis from at least two independent assays performed is shown. The gel images were cropped to show selected samples. M, DNA ladder; Un, undigested; 1x, monomer; 2x, dimer; 3x, trimer; 4x, tetramer. (B) Metagenome plot showing cohesin enrichment ± 1000 bp from the transcription start site (TSS) in WT and *rad30Δ* cells ± DSB induction in $G_2$/M phase. The samples were first normalized to their respective input and then the values were scaled to the maximum value of the plot. (C) The data from (B) plotted relative to the WT-DSB sample. After normalizing to the input, all samples were also normalized to WT-DSB sample to visualize the changes between the WT and *rad30Δ* cells. (D) Metagenome plot showing cohesin distribution 1000 bp downstream and 100 bp upstream from the transcription end site (TES) in WT and *rad30Δ* cells ± DSB induction in $G_2$/M phase. Plotted as in (B). (E) As in (C), except plotting cohesin distribution around the TES according to (D). (F) Monitoring activation of the DNA damage checkpoint (phosphorylation of Rad53) after DSB induction with western blot. Galactose was added into the $G_2$/M arrested cell cultures to induce $P_{GAL}$-*HO* break induction for 1- or 1.5-hour, denoted as 1G or 1.5G. Sample collected from $G_2$/M arrested WT cells, treated with phleomycin (final 15 μg/ml) for 1.5 hours was included as positive control (PC). Cdc11 was used as loading control. M, protein marker. (G) Monitoring activation of DNA damage checkpoint during DSB recovery. DSBs were induced for 1- or 1.5-hour, as in (F). The cells were then allowed to recover in YEP media supplemented with glucose and benomyl for another 1.5 hour (1.5 R) at 35˚C, to mimic the damage-induced cohesion assay. 1G, 1.5G, PC, M as in (F). Cdc11 was used as loading control. (H) FACS analyses of cell cycle progression in WT and *rad30Δ* cells, at indicated time points after release into YEP media supplemented with glucose to recover from DSB induction. Samples without DSBs were included as control. B, benomyl; R, recovery. (I) *ECO1* gene expression in $G_2$/M arrested WT and *rad30Δ* cells ± $P_{GAL}$-*HO* (left) and ± γ-irradiation (right). The relative gene expression was measured by RT-qPCR. FBA1 was used as a reference gene for the ± $P_{GAL}$-*HO* samples. Error bars indicate the mean ± STDEV of two independent experiments.
(TIFF)

**S5 Fig. The method and related control experiments for a typical damage-induced cohesion assay.** (A) Damage-induced cohesion assay performed with *GAL* induced DSBs on chromosome III ($P_{GAL}$-*HO*). Strains harboring the temperature sensitive *smc1-259* allele are arrested in $G_2$/M by addition of benomyl ('B'). Galactose is then added for expression of ectopic $P_{GAL}$-*SMC1-MYC* (Smc1 WT) and induction of DSBs, for 1-hour. The temperature is then raised to 35˚C, restrictive to the *smc1-259* allele, for disruption of S-phase cohesion (blue rings). The Tet-O/TetR-GFP system (green dots) is used to monitor damage-induced cohesion (red rings) on chr. V. Chr., chromosome; III, three; V, five. B1 and 2 indicate replacement of media with freshly prepared benomyl. (B) FACS analysis to confirm $G_2$/M arrest during the time course of a typical damage-induced cohesion assay. 3B, 3-hour benomyl arrest. (C) PFGE analysis to detect DSB induction on chromosome III. 1, $G_2$/M arrest; 2, 1-hour *GAL*-induction ($P_{GAL}$-*HO* and $P_{GAL}$-*SMC1-MYC*). (D) Western blot to check expression of the *GAL* promoter driven ectopic Smc1-Myc protein. G2, $G_2$/M arrest; 1G, 1-hour *GAL*-induction as in (C).

Cdc11 was used as loading control. M, protein marker.
(TIFF)

**S6 Fig. Damage-induced cohesion assay performed with γ-irradiation and the maintenance of sister chromatid cohesion in *htz1Δ* cells.** (A) Damage-induced cohesion assay performed with γ-irradiation. Formation of damage-induced cohesion is monitored on chr. V with the same Tet-O/TetR-GFP system, as in S5A Fig, with slight differences in the experimental procedure. Strains with *smc1-259* background are arrested in $G_2$/M by addition of benomyl ('B'), expression of ectopic $P_{GAL}$-*SMC1-MYC* (Smc1 WT) is then induced by addition of galactose. The cells are subsequently pelleted, resuspended in 1X PBS supplemented with benomyl. The resuspension is split in one half for irradiation, and half as non-irradiated control. After irradiation, both ± irradiated cells are recovering in YEP media supplemented with galactose and benomyl. Subsequently, the media is changed to YEP containing glucose and benomyl, and the temperature raised to 35˚C, to monitor formation of damage-induced cohesion. (B) Damage-induced cohesion assay of the *hir1Δ* mutant in response to γ-irradiation, performed as depicted in (A). Means ± STDEV from two independent experiments are shown. For each experiment, two-hundred cells were counted for each time point. (C) Sister chromatid cohesion maintenance of the *htz1Δ* mutant under prolonged $G_2$/M arrest. The cells were initially synchronized in $G_1$ by α-factor in YEP media containing galactose. Expression of $P_{GAL}$-*CDC20* was then shut off by switching the carbon source to glucose (YEPD), which resulted in the subsequent prolonged $G_2$/M arrest as monitored by FACS (left panel). Sister chromatid separation was monitored at the *URA3* locus on Chr. V by the TetO/TetR-GFP system. Means ± STDEV from three independent experiments are shown (right panel). A *rad61Δ* mutant with known high sister separation under prolonged $G_2$/M arrest was included as control. Parts of the results from the same experiments were previously published [28]. Chr., chromosome.
(TIFF)

**S7 Fig. Control experiments for the Rpb1-anchor away method.** (A-C) ChIP-qPCR analyses to determine chromatin association of Rpb1 at promoters and 3'-ends of selected genes, in $G_2$/M arrested cells after DSB induction. The same genes as in Fig 7C–7E were analyzed. Error bars indicate the mean ± STDEV of three independent experiments. n, low-binding control (n2 in Fig 1B). (D) Representative *in situ* immunofluorescence images for samples collected from the damage-induced cohesion assays in Fig 7F. The cells were stained with anti-Rpb1 and then counterstained with DAPI. $t_0$, the time point before splitting the culture for addition of rapamycin (RAP); 1h RAP, 1-hour after ± rapamycin; 2nd step, the secondary antibody alone as control. (E) Fold reduction of selected genes after 1-hour rapamycin treatment, measured by RT-qPCR. The $2^{-\Delta Ct}$ values of untreated samples were set as 1. (F) Western blot to monitor early DNA damage response, as indicated by H2AS129-phosphorylation. RAP, rapamycin; R, recovery; IR, γ-irradiation (250 Gy); M, protein marker. Cdc11 was used as loading control. (G) Western blot to check expression of the ectopic Smc1-Myc, driven by the *GAL* promoter. G2, $G_2$/M arrest; 1G, 1-hour *GAL*-induction. RAP, M, Cdc11 as in (F).
(TIFF)

**S8 Fig. A summary of the main results.** In $G_2$/M arrested WT cells, genes belonging to the positive transcription regulation and chromatin assembly pathways are enriched compared to *rad30Δ* cells. Reduced chromatin assembly in *rad30Δ* cells results in less dynamic chromatin, indicated by additional nucleosomes. Deregulated transcription and sensitivity to elongation inhibitors in *rad30Δ* cells are indicated by thin arrows over the TSS and ORF. Histone exchange between H3 and the post-translationally modified H3 (H3K56Ac) at promoter

regions is reduced in the *hir1Δ* mutant, while histone exchange of H2A.Z for H2A predominantly at the +1 nucleosome is prevented in the *htz1Δ* mutant, hampering transcriptional regulation. Both mutants were deficient in damage-induced cohesion. In contrast, deletion of *SET2* compensated for reduced transcriptional capacity of the *rad30Δ* mutant, and suppressed the lack of damage-induced cohesion in *rad30Δ* cells. Taken together, histone exchange during transcription may facilitate formation of damage-induced cohesion. Transcriptional regulation is perturbed in *rad30Δ* cells, and this appeared to have a consequence on generation of damage-induced cohesion. Cells with a single green dot indicates established damage-induced cohesion while cells with two dots indicates lack of damage-induced cohesion. Since Polη may play an indirect role in transcription, recruitment of Polη to the promoter region is indicated with a dashed double ended arrow. ORF, open reading frame.
(TIFF)

**S1 Data. Differential gene expression analysis.**
(XLSX)

**S2 Data. GSEA summary *rad30Δ* G2 versus WT G2.**
(XLSX)

**S3 Data. GSEA summary DSB versus G2.**
(XLSX)

**S4 Data. Numerical data of graphs.**
(XLSX)

**S5 Data. Summary of statistical analyses.**
(XLSX)

**S1 Table. Strains used in this study.**
(DOCX)

**S2 Table. Information on used primary antibodies.**
(DOCX)

**S3 Table. Primers used in ChIP-qPCR.**
(DOCX)

**S4 Table. Primers used in RT-qPCR.**
(DOCX)

## Acknowledgments

We thank K. Shirahige and Y. Katou for support and technical assistance during collection of the previously published ChIP-Seq data. We thank C. Björkegren for sharing strains and plasmids, and for critical reading of the manuscript. We thank K. Jeppsson and V. Kuzin for advice on library preparation and data analysis for Polη-ChIP-seq. We thank Claudia Kutter for technical expertise and access to the sequencing machine.

## Author Contributions

**Conceptualization:** Pei-Shang Wu, Lena Ström.

**Data curation:** Pei-Shang Wu, Jan Grosser, Donald P. Cameron.

**Formal analysis:** Pei-Shang Wu, Jan Grosser, Donald P. Cameron.

**Funding acquisition:** Laura Baranello, Lena Ström.

**Investigation:** Pei-Shang Wu, Lena Ström.

**Methodology:** Pei-Shang Wu.

**Project administration:** Lena Ström.

**Resources:** Laura Baranello, Lena Ström.

**Software:** Jan Grosser, Donald P. Cameron.

**Supervision:** Laura Baranello, Lena Ström.

**Validation:** Pei-Shang Wu, Lena Ström.

**Visualization:** Pei-Shang Wu, Jan Grosser, Donald P. Cameron.

**Writing – original draft:** Pei-Shang Wu.

**Writing – review & editing:** Pei-Shang Wu, Jan Grosser, Donald P. Cameron, Laura Baranello, Lena Ström.

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
