## [Decision Letter · Decision Letter 0]

13 Apr 2021

Dear Dr Ström,

Thank you very much for submitting your Research Article entitled 'The assisting role of Pol η in transcription facilitates formation of damage-induced cohesion' to PLOS Genetics.

The manuscript was fully evaluated at the editorial level and by independent peer reviewers. The reviewers appreciated the attention to an important problem, but raised some substantial concerns about the current manuscript. Based on the reviews, we will not be able to accept this version of the manuscript, but we would be willing to review a much-revised version. Reviewers 1 and 3 raise a number of major concerns that would need to be addressed experimentally.  We cannot, of course, promise publication at that time.

If you decide to revise the manuscript for further consideration at PLOS Genetics, please aim to resubmit within the next 60 days, unless it will take extra time to address the concerns of the reviewers, in which case we would appreciate an expected resubmission date by email to plosgenetics@plos.org.

[LINK]

We are sorry that we cannot be more positive about your manuscript at this stage. Please do not hesitate to contact us if you have any concerns or questions.

Yours sincerely,

Lorraine S. Symington

Associate Editor

PLOS Genetics

Gregory P. Copenhaver

Editor-in-Chief

PLOS Genetics

Reviewer's Responses to Questions

**Comments to the Authors:**

Reviewer #1: Cohesin binds around DSBs and is important for repair. Interestingly, DSBs trigger a genome-wide reinforcement of cohesion. This manuscript seeks to build on previous work implicating RAD30 in the genome-wide process but not DSB-local cohesion. They use an engineered budding yeast strain with a ts mutation in smc1, engineered to make a gal inducible HO break on chromosome 3 and monitor cohesion on chromosome 5. In a few cases gamma irradiation is used instead of an HO break. The authors investigate how RAD30 contributes to transcription as a mechanism by which it generates genome-wide cohesion. The hypothesis put forward is that RAD30 facilitates DI-cohesion via its role in the transcriptional response to DSBs. The data in support of this hypothesis needs to be stronger to warrant publication.

Summary of Figures:

In figure 1 they demonstrate Rad30 mutant strains are sensitive to transcriptional inhibitors and Rpb1 ChIP shows it has reduced localization to several genes despite normal protein levels. In figure 2 they demonstrate that gene expression is disrupted in the RAD30 deletion strain under normal growth conditions and the transcriptional response to a DSB is attenuated. The authors argue that this attenuated transcriptional program could underlie the lack of genome-wide damage-induced cohesion. However, by this logic, any mutation that reduces the transcriptional response to a DSB would have this phenotype-this is a point that should be further probed experimentally. In Figure 3, they show that rad30 mutants have more cohesin specifically at the TSS, with no difference in the TES, and irrespective of DSB induction. The relationship of this information to the main point could be more clearly stated. Figure 4 shows the gene expression signatures, further analyzing the data from Figure 2. In the next section of the manuscript the authors switch to genetic approaches to probe chromatin proteins that influence DI-cohesion, based in part on the gene expression data in the rad30 mutant. In figure 5 they show that deletion of HIR1 partially rescues DI cohesion in the Rad30 mutant background whereas histone deletions have no DI-cohesion defects and do not rescue. In figure 6 they test the effect of deleting the Htz1 histone variant. They cannot use the HO system so they induce breaks with radiation. They show that Htz1 levels are higher by ChIP following a break in the rad30 mutant. In Figure 7 they show Set2 can partially rescue DI-cohesion in the rad30 background. They also show ChIP data for Rpb1 but I can’t figure out what I am supposed to take away from the ChIP experiment. In Figure S7 they attempt to integrate their findings into a working model. I appreciate the effort to show a working model but it needs more work. In summary it seems that Hir1 and Htz1 affect histone turnover and DI-cohesion is defective. Do they mean to imply that Rad30 affects histone turnover directly? Indirectly? Why does altered elongation with set 2 deletion rescue the rad30 phenotype? If DI-cohesion is defective in rad30 due to an attenuated transcriptional response to a DSB, does set2 deletion rescue by rescuing the transcriptional response in rad30?

For me, the major question presented in the introduction, which is whether transcription contributes to genome-wide DI-cohesion, is an interesting one. But the manuscript does not meet the bar to answer this question, or even partially answer it with respect to RAD30 function. Results are not sufficiently integrated with each other, making it difficult to understand how each experiment addresses the overarching question of how transcription contributes to genome-wide DI cohesion. The data provided is not sufficient to support the hypothesis that Rad30 deletion blocks the formation of damage induced cohesion via altering the transcriptional program. Furthermore, its not clear whether the effect is direct or indirect. I have suggested three experiments below that would potentially begin to further examine the hypothesis, but depending on the outcome, more experiments may be warranted. This includes ChIP of Rad30, to be integrated with gene expression data, inhibiting transcription and examining DI-cohesion, and further experimentation to understand how deletion of Set2 rescues Rad30.

Major concerns and suggestions for how to improve the manuscript

1. Are the promoter types that are associated with differentially expressed genes in the rad30 mutant promoters at which Rad30 binds? This information would help argue for a direct role for Rad30 in the transcription of these genes, and would help support the model.

2. It’s a shame thiolutin induced a DNA damage response on its own and prevented them from testing DI-cohesion when transcription was inhibited, as this could help support the model. Could the authors find a different way to do this experiment? What about ActD treatment?

3. It’s not clear to me why the authors decided to examine the specific chromatin mutants chosen (Hir1, Htz1, Set2). Were these genes highly dysregulated in the gene expression data or chosen some other way? To me the manuscript seems to jumps around from one mutant to another without much logic and just when we start to understand how one mutation is acting they move to another. A more in depth analysis would be helpful to arrive as a more mechanistic understanding, perhaps focusing on how deletion of set2 rescues rad30. One experiment that could be done is examine whether the double mutant has a transcriptional profile that is more wild-type relative to the rad30 mutant profile, which would support the model.

Minor concerns

4. Please add relevant statistics to Figure 2 D and E to demonstrate the change from WT.

5. I’m not sure the results regarding short genes in figure 2F are relevant to the overall message.

6. The cohesion defect for RAD30 in part 5A is much larger than in part C and so a partial rescue would be much harder to observe.

7. For Figure 6, the authors should check Htz1 proteins levels by western to help with interpretation of the ChIP. Are the levels staying the same or going up in the rad30 mutant?

8. The authors have used a graphical image of budding yeast cartoon to diagram cohesion but it doesn’t print well in my copy. It needs a black outline.

Reviewer #2: In their manuscript PGENETICS-D-20-00364 Wu and colleagues attempts to define the contribution of the yeast TLS DNA Pol eta to DNA damage-induced cohesion formation, a phenomenon the same group reported in 2013 in Plos Genetics. Here, they conclude that Pol eta affects damage-induced cohesion through its novel role in transcription. First they confirm the role of Pol eta in transcription, previously reported by others, using RNA seq., establishing that a few hundred genes are up or downregulated in the absence of the RAD30 gene coding for Pol eta. The reduced association of RNA PolII at promoters and coding regions in rad30 delta detected by ChIP-qPCR agrees with a transcriptional regulatory role of Pol eta. Intriguingly, cohesin association at promoters is increased in rad30 delta, still damaged-induced cohesion is impaired. They found similar deficiency when deleting HIR1 needed for H3 exchange for transcription activation, and suppression of the rad30 delta transcriptional and cohesin formation defects by deletion of the SET2 histone methyl transferase repressing histone exchange during transcription elongation. Based on these they conclude that deregulated transcription in rad30 delta, which perturbs dynamic nucleosome assembly affects formation of damage-induced cohesion.

The topic of the paper DNA damage induced cohesion formation is highly interesting, and the results represent substantial contribution. The experiments are carefully executed with all the necessary controls, and the paper is clearly written.

I have one concern regarding the experimental repetitions. The authors state that the results indicate the mean ± STDEV of at least two independent experiments, in almost all figure legends. Two experiments are not enough for solid statistics and conclusions, especially when STDEV is high. They should indicate the exact numbers of biological and technical replicates for each experiment, and perform additional repetitions if necessary.

Reviewer #3: This manuscript investigates the potential links between transcription and damage-induced cohesion through a common role of Rad30, the DNA polymerase eta involved in translesion synthesis. The authors first tested the toxicity of transcription inhibitors in rad30D strains and observed increased sensitivity as well as a decrease in the amount chromatin bound PolII. Similar observations were obtained with the DI-cohesion mutant of Rad30, Rad30-S14A. The authors observed deregulation of transcription profiles for Rad30D in the presence and absence of DNA damage hence suggesting a role for Rad30D in general transcription regulation. Moreover, the levels of cohesin binding at TSS in Rad30D cells were increased and this led the authors investigate the link between DI-cohesion and transcription at promoter sites. To this aim they used hir1D and htz1D mutations, which disrupt histone assembly at gene promoters. They found that both mutants decrease the ability of cells to establish DI-cohesion, thus concluding that histone exchange disruption at promoters causes defects in damage-induced cohesion. Finally, the authors show that deletion of histone methyltransferase set2 suppresses the DI-cohesion defects of Rad30. The authors conclude from this data that transcription influences formation of damage-induced cohesion, and that Rad30 is required for damage-induced cohesion because it facilitates transcription.

Overall, this manuscript makes some interesting observations and indeed makes some weak links between Rad30, transcription and DI-cohesion. It has been previously shown that cohesin loading requires nucleosome free regions and therefore changes in transcription or nucleosome assembly/chromatin at promoters (in hir1D and hzt1D) are indeed expected to have an impact on DI-cohesion. The role of Rad30 is however intriguing and although the changes in transcription in Rad30D cells could indeed explain some effects, the links are very vague and often do not offer a clear cut explanation of how Rad30 might be either involved in transcription regulation or facilitate DI-cohesion (by altering transcription), as the authors only offer some genetic interactions. Also, the use of Rad30D mutants might not be the best experimental system. The authors should use Rad30 auxin degron strains where they could remove Rad30 upon DNA damage and avoid the background effect of the deletion.

To support their claims it would be important for the authors to illustrate their points using a handful of specific genome sites where they can investigate changes in transcription and cohesin loading (or cohesion establishment) upon damage in a manner dependent on Rad30 (using an auxin degron approach). This would significantly strengthen their study.

**Have all data underlying the figures and results presented in the manuscript been provided?**

Reviewer #1: Yes

Reviewer #2: Yes

Reviewer #3: Yes

PLOS authors have the option to publish the peer review history of their article (what does this mean?). If published, this will include your full peer review and any attached files.

Reviewer #1: No

Reviewer #2: No

Reviewer #3: No

---

## [Decision Letter · Decision Letter 1]

26 Jul 2021

Dear Dr Ström,

Thank you very much for submitting your Research Article entitled 'Transcriptional deregulation driven by absence of Polη brings negative impact on damage-induced cohesion' to PLOS Genetics.

The manuscript was fully evaluated at the editorial level and by independent peer reviewers. While two of the reviewers were satisfied with the revision, reviewer 1 identified some concerns that we ask you address in a revised manuscript. In particular, it would be important to confirm that addition of rapamycin does not impact DI-cohesion in WT cells. You might also consider the alternative title for the manuscript suggested by reviewer 1.

We therefore ask you to modify the manuscript according to the review recommendations. Your revisions should address the specific points made by each reviewer.

[LINK]

Yours sincerely,

Lorraine S. Symington

Associate Editor

PLOS Genetics

Gregory P. Copenhaver

Editor-in-Chief

PLOS Genetics

Reviewer's Responses to Questions

**Comments to the Authors:**

Reviewer #1: I appreciate that the authors executed and added a number of additional experiments to address reviewers’ concerns. The overarching question is quite interesting. Most of the individual experiments have proper controls, repeats, and quantification. However, I do not think the authors have reached the bar. The third reviewer commented in the first round of reviews that the link between Rad30, transcription, chromatin, and DI-cohesion is vague, and I think this problem persists in the revised version, although it has improved some in terms of knitting together a story. The constitutive genetic deletion approach makes it difficult to ascertain how the mutations are affecting transcription in the short term and long term, and if the effects are direct versus indirect, and how this corresponds with the effect on the loss of damage induced cohesion. All the null backgrounds allow for transcription to persist, so it must be something more specific about the transcriptional process that leads to loss of damage-induced cohesion. The concerns below are not an exhaustive list of weaknesses. Even if these issues could be addressed, the manuscript may still not meet the bar.

1. New evidence is presented suggesting that the effect of Rad30 deletion is probably indirect as the binding does not correlate with transcriptional changes. This correlation between binding and transcription, or lack thereof, would have been much stronger if the experiments were performed genome-wide and not at a handful of genes.

2. It’s not clear to me if loss of Rad30 changes histone turnover or histone acetylation. It seems premature to conclude that Rad30 perturbs transcription through a similar mechanism to either Htz1 or Hir1, based on the data included.

3. I appreciate that the authors added the experiment to induce the removal of Rpb1 to impact transcription. This is potentially a very important experiment. As a control the authors should validate that the addition of rapamycin alone does not impact damage-induced cohesion in a WT background, since rapamycin will impact the transcriptional program even in the absence of the anchors away allele of Rpb1.

4. Rad30 seems to have its effect on genes with a specific group of promoters. How should I think about the effect on specific promoters relative to damage induced cohesion? Does the actual gene expression program have any bearing on DI-cohesion? The focus on the genes up and down regulated in the null background, which is a long term effect, and the promoter types bound, confuses the issue of the effect on DI-cohesion for me. Stripping away extraneous information may help convey the message better.

Minor concerns.

1. The adjusted title is poorly formulated. The authors should state their findings in the positive. A preferable title would be something like “Transcription-related chromatin features facilitate damage-induced cohesion”

2. The writing style in the manuscript is improved from the original but it is still often difficult to follow the experiments and the logic.

3. The reply to reviewers has, at some points, a similar problem with clarity.

4. The quality/size of the figures was so poor in my pdf that I had to download figures one by one, and enlarge them.

5. Given comments 2-4, the reviewer’s work is much more challenging than for an average review assignment. The authors and journal should strive to provide a manuscript in a more easily reviewable state.

Reviewer #2: I accept the modifications.

Reviewer #3: The authors have addressed, or explained, most of my concerns, I find the revised version improved. I am now supportive of publication.

**Have all data underlying the figures and results presented in the manuscript been provided?**

Reviewer #1: Yes

Reviewer #2: Yes

Reviewer #3: Yes

PLOS authors have the option to publish the peer review history of their article (what does this mean?). If published, this will include your full peer review and any attached files.

Reviewer #1: No

Reviewer #2: No

Reviewer #3: No

---

## [Editor Report · Decision Letter 2]

5 Aug 2021

Dear Dr Ström,

We are pleased to inform you that your manuscript entitled "Deficiency of Polη in Saccharomyces cerevisiae reveals the impact of transcription on damage-induced cohesion" has been editorially accepted for publication in PLOS Genetics. Congratulations!

Yours sincerely,

Lorraine S. Symington

Associate Editor

PLOS Genetics

Gregory P. Copenhaver

Editor-in-Chief

PLOS Genetics

Comments from the reviewers (if applicable):

**Data Deposition**

http://datadryad.org/submit?journalID=pgenetics&manu=PGENETICS-D-21-00364R2

**Press Queries**

---

## [Editor Report · Acceptance letter]

4 Sep 2021

PGENETICS-D-21-00364R2 

Deficiency of Polη in *Saccharomyces cerevisiae* reveals the impact of transcription on damage-induced cohesion   

Dear Dr Ström, 

We are pleased to inform you that your manuscript entitled "Deficiency of Polη in *Saccharomyces cerevisiae* reveals the impact of transcription on damage-induced cohesion  " has been formally accepted for publication in PLOS Genetics! Your manuscript is now with our production department and you will be notified of the publication date in due course.

With kind regards,

Andrea Szabo

PLOS Genetics

On behalf of:
